# QSAR Studies, Synthesis, and Biological Evaluation of New Pyrimido-Isoquinolin-Quinone Derivatives against Methicillin-Resistant *Staphylococcus aureus*

**DOI:** 10.3390/ph16111621

**Published:** 2023-11-17

**Authors:** Juan Andrades-Lagos, Javier Campanini-Salinas, Gianfranco Sabadini, Victor Andrade, Jaime Mella, David Vásquez-Velásquez

**Affiliations:** 1Facultad de Medicina y Ciencia, Universidad San Sebastián, Santiago 7510157, Chile; juan.andrades@uss.cl; 2Drug Development Laboratory, Faculty of Chemical and Pharmaceutical, Sciences, Universidad de Chile, Santiago 8380492, Chile; 3Facultad de Medicina y Ciencia, Universidad San Sebastián, Puerto Montt 5501842, Chile; 4Instituto de Química y Bioquímica, Facultad de Ciencias, Universidad de Valparaíso, Av. Gran Bretaña 1111, Valparaíso 2360102, Chile; gianfranco.sabadini@postgrado.uv.cl; 5Centro de Investigación Farmacopea Chilena, Facultad de Farmacia, Universidad de Valparaíso, Av. Gran Bretaña 1093, Valparaíso 2360102, Chile; 6Laboratory of Neuroscience and Functional Medicine, International Center for Biomedicine, Faculty of Sciences, University of Chile, Santiago 7800003, Chile; victormanuel.andradefuentes@uk-koeln.de; 7Division of Neurogenetics and Molecular Psychiatry, Department of Psychiatry and Psychotherapy, Medical Faculty, University of Cologne, 50923 Köln, Germany; 8Department of Neurodegenerative Diseases and Geriatric Psychiatry, University Hospital Bonn, 53127 Bonn, Germany

**Keywords:** antibacterial agents, drug discovery, quinone antibiotics, structure-activity relationships, 3D-QSAR, CoMFA, CoMSIA, MRSA; methicillin-resistant *Staphylococcus aureus*, antimicrobial resistance

## Abstract

According to the WHO, antimicrobial resistance is among the top 10 threats to global health. Due to increased resistance rates, an increase in the mortality and morbidity of patients has been observed, with projections of more than 10 million deaths associated with infections caused by antibacterial resistant microorganisms. Our research group has developed a new family of pyrimido-isoquinolin-quinones showing antibacterial activities against multidrug-resistant *Staphylococcus aureus*. We have developed 3D-QSAR CoMFA and CoMSIA studies (r^2^ = 0.938; 0.895), from which 13 new derivatives were designed and synthesized. The compounds were tested in antibacterial assays against methicillin-resistant *Staphylococcus aureus* and other bacterial pathogens. There were 12 synthesized compounds active against Gram-positive pathogens in concentrations ranging from 2 to 32 µg/mL. The antibacterial activity of the derivatives is explained by the steric, electronic, and hydrogen-bond acceptor properties of the compounds.

## 1. Introduction

Antibacterial resistance is a growing global health threat [1]. The World Health Organization (WHO) has warned that we are on the brink of a post-antibiotic era, where common infections could once again be deadly [2]. Indeed, antimicrobial resistance is considered one of the top 10 threats to global health [3]. While this is a natural process, the increase in antimicrobial resistance is due to the exposure of bacteria to antibacterial drugs and the subsequent spread of these bacteria which exhibit various resistance mechanisms, accelerated by exposure to significant amounts and/or prolonged durations of antibiotics in patients or in the environment [4]. This is because bacteria are becoming increasingly resistant to the antibiotics that we have used to treat them for decades [5].

Projections indicate that if there is no change, there will be an increase in deaths each year in the number of hospitalizations and in economic costs associated with antimicrobial resistance [6,7,8]. For example, in the USA, it is estimated that each year, there will be almost 3 million infections caused by antibiotic-resistant bacteria or fungi, with at least 36,000 deaths [9]. In 2016, the World Bank report showed that this problem will be associated with a loss of between 1.1 and 3.8% of gross domestic product in different countries, and up to 5% in low-income countries. As a consequence, the number of people living in poverty and healthcare costs are projected to increase [10].

The most problematic bacteria for global public health were grouped by Lois B. Rice in 2008, under the acronym “No ESKAPE” [11]. The pathogens represented are *Enterococcus faecium*, *Staphylococcus aureus*, *Klebsiella pneumoniae*, *Acinetobacter baumanii*, *Pseudomonas aeruginosa*, and *Enterobacter* spp. The following year, Peterson suggested modifying the first acronym to “no ESCAPE” by including *Clostridioides difficile* and replacing *Enterobacter* spp. with Enterobacteriaceae [12]. In this context, the WHO has proposed a global action plan on antimicrobial resistance to stimulate the research and development of new antibacterial drugs, which is essential to combat the rise in antibacterial resistance [13]. Despite this situation, in recent decades, the discovery of new antibacterial drugs has slowed dramatically [14,15,16]. Also, when analyzing the antibiotics introduced in the last two decades, most of them are related to previously introduced antibiotics and do not offer an innovative mechanism of action [14].

To this end, the WHO in 2017 published a list of priority microorganisms to guide the research and development of new antibacterial drugs. At priority 1 (critical) are Gram-negative bacteria such as carbapenem-resistant *Acinetobacter baumannii*, carbapenem-resistant *Pseudomonas aeruginosam*, and ESBL-producing carbapenem-resistant *Enterobacteriaceae* [17]. At priority 2 (high), a number of bacteria are found, including vancomycin-resistant *Enterococcus faecium* and methicillin-resistant *Staphylococcus aureus* with intermediate sensitivity or resistance to vancomycin [18]. In particular, methicillin-resistant *Staphylococcus aureus* (MRSA) is the main cause of *Staphylococcus aureus*-associated mortality, which generates infections such as bacteremia, causing millions of deaths per year and with estimated economic losses of around 14 billion dollars [18,19,20].

On the other hand, the traditional drug discovery process is slow and expensive. It can take many years and millions of dollars to bring a new antibiotic onto the market [21]. This is why there is a growing interest in using computational chemistry methods to search for new antibacterial drugs [22]. In response to this problem, we have reported the discovery of a new family of pyrimido-isoquinolin-quinone antibiotics [23]. These derivatives have potent antibacterial activity against Gram-positive microorganisms like *Enterococcus faecium* and *Staphylococcus aureus* [24]. Recently, we reported the qualitative structure–activity relationship of this novel family of antibiotic compounds [25].

Quantitative structure–activity relationships (QSAR) methods can be used to formulate equations in order to design new antibacterial drugs [26]. For example, 3D-QSAR CoMFA (comparative molecular field analysis) [27] and CoMSIA (comparative analysis of molecular similarity index) [28] represent very useful methodologies to design new compounds and predict their antibacterial activity previous to their synthesis. Among the advantages of these methods are that (a) they do not require knowledge of the structure of the target [29]; (b) they allow us to understand how the three-dimensional properties (steric, electrostatic, hydrophobic, hydrogen bond donor, and acceptor potentials) contribute to biological activity [30]; and (c) they save time and resources by allowing the design of compounds and predicting their biological activity. Today, it is even possible to carry out processing on cloud servers [31]. The limitations of these methods are that they are highly dependent on molecular alignment, so the presence of a common core of the compounds is usually an important requirement [32]. Another limitation is that unlike receptor-based studies, it is required to experimentally measure the biological activity of the compounds, to use it as a dependent variable [33].

The use of computational chemistry methods has already led to the discovery of a number of new antibacterial drugs [34]. For example, the antibiotic teixobactin was discovered using computational chemistry methods like structure-based drug design [35]. To date, there are no QSAR studies on pyrimido-isoquinolin-quinones with anti-MRSA activity. In this study, using CoMFA and CoMSIA methods, we identified structural changes in the steric and electronic properties of the compounds that can improve the antibacterial activity of new derivatives. In the present work, we report the first 3D-QSAR study of a series of 44 pyrimido-isoquinolin-quinone compounds with the aim of obtaining new compounds active against methicillin-resistant *Staphylococcus aureus*. Our primary objective was to formulate models that would explain the structure–activity relationship of our compounds in terms of three-dimensional variables, and that would allow the design and synthesis of new antibacterial compounds.

## 2. Results and Discussion

### 2.1. CoMFA/CoMSIA Studies

Given the urgent need to develop new compounds active against methicillin-resistant *Staphylococcus aureus*, we used previously published compounds synthesized by our group (compounds 1–32) to develop QSAR models based on CoMFA/CoMSIA studies against methicillin-resistant *Staphylococcus aureus* (ATCC^®^ 43300) [23,25]. The best CoMFA and CoMSIA models were sought out using a sequential search of field combinations (see Appendix A). The steric, electrostatic, hydrophobic, hydrogen bond acceptor, and hydrogen bond donor fields were the independent variables which were correlated with the biological activities. The best models are presented in Table 1. The selection criterion for the best model was the highest possible q^2^ value. The best CoMFA model considered only the steric contribution to biological activity, while the best CoMSIA model considered steric (26.9%), electrostatic (50.4%), and hydrogen bond acceptor potential (22.7%). Both models had a low number of components (N = 5) and adequate q^2^ (0.660 and 0.596) and r^2^ values (0.938 and 0.895). To achieve more thorough validation, the Y-randomization method was employed. In this method, the biological activity is randomized, and the q^2^ value is recalculated for a total of 10 new randomized models (Table 2). If low or negative q^2^ values are obtained, it is concluded that the initially obtained models are not the result of random correlation. In the 10 new randomized models, q^2^ values were less than 0.11, with the majority being negative. The average q^2^ values for CoMFA and CoMSIA were −0.193 and −0.161, respectively, demonstrating the robustness of the proposed models.

The 32 compounds studied were divided into a training (22 compounds, 70%) and a test set (10 compounds, 30%) (Table 3). In CoMFA, a total of 15 compounds had negative residuals (Residual = pMIC_Exp_ − pMIC_Pred_), and 17 had positive residuals. Meanwhile, in CoMSIA, a total of 18 compounds had negative residuals, and 14 had positive residuals. Therefore, both models exhibit a balanced predictive capability without showing tendencies to overestimate or underestimate activities. Figure 1 presents the distribution graphs of experimental activity versus predictive activity for both models. A good distribution of values along the *y = x* line can be observed, spanning approximately two logarithmic units of antibacterial activity. Compound **2** exhibited the highest deviation, with a residual of −1.29 in CoMFA and −0.88 in CoMSIA. The biological activity value of compound **2** was overestimated by the model predictions. This compound features a methyl group in the *ortho* position of the benzene ring, which could generate a specific dihedral angle between this ring and the rest of the quinone. This conformation could adversely affect its ability to establish π-stacking interactions with the target. The information obtained from contour maps is discussed below.

### 2.2. CoMFA Contour Maps

In the steric contour map of CoMFA (Figure 2), green and yellow polyhedral can be observed around the aniline and thiol fragments of the compounds. A green color means that the insertion of bulky substituents would be favorable for activity, while yellow indicates that the insertion of compact substituents would be favorable. The presence of NH or sulfur markedly affects the bond angle. The insertion of bulky substituents is favorable in both cases. Green polyhedra are observed near the *meta* and *para* positions of compound 23 (which has an NH linker). Additionally, a green polyhedron is observed near the *ortho* position of derivative 4 (with a sulfur linker). The yellow polyhedron further away from the benzene rings suggests that increasing the volume has a limit. For this reason, we propose the insertion of short bulky groups, while long and branched chains should be avoided. In fact, in the studied series, the most active compounds, 4 (pMIC = 5.6869) and 24 (pMIC = 5.9725), feature a bulky bromine atom in the *ortho* and *para* positions, respectively. On the other hand, less active compounds like 25 (pMIC = 4.9701) and 26 (pMIC = 4.9861) have alkyl chains with three and four carbon atoms, respectively. To gain further insight into the structure–activity relationship, we conducted a CoMSIA study, the results of which are presented below.

### 2.3. CoMSIA Contour Maps

In the CoMSIA steric contour map (Figure 3), a small green polyhedron is observed in the ethyl chain. This suggests that in this position, it is preferable to have an atom other than hydrogen. Therefore, we suggest maintaining a methyl or ethyl group connected to the ring. Given the small size of the green polyhedron, the use of larger groups like propyl, butyl, or even cycles should be further investigated to draw more significant conclusions about the volume effect in this position. Additionally, a green polyhedron is observed at the *para* position of the benzene ring, similar to what was observed in CoMFA. This reinforces the idea of inserting voluminous atoms in these positions. Interestingly, two yellow polyhedra are observed near the *meta* position of the benzene ring in compounds **4** and **24**. This seems to indicate that a significant steric factor is responsible for the activity of these compounds, as antibacterial activity would be sensitive to the position of the substituents in this ring.

Furthermore, the electrostatic contour map (Figure 3) shows in red that electron-rich atoms would be favorable, while blue indicates that electron-deficient atoms would be favorable. A red polyhedron is observed right above the bromine atom of compound **4**. On the other hand, the blue polyhedron is positioned over the benzene ring. Interpreting both polyhedra together, we can conclude that the presence of halogens or electron-attracting groups like nitro, nitrile, or carbonyl in the *ortho* and *para* positions would be favorable for biological activity. From a potential mechanism of action perspective, we can postulate that the presence of π-stacking interactions with electron-rich residues such as phenylalanine or tyrosine could play a key role in the activity of these compounds. Additionally, the presence of halogens could be crucial in the potential formation of halogen bonds with the target.

Finally, CoMSIA also provided relevant information regarding the hydrogen bond acceptor capacity of the compounds. In Figure 4, a large magenta polyhedron is observed, indicating that the presence of hydrogen bond acceptor atoms is favorable for activity. A small-sized polyhedron is positioned over the sulfur atom of compound **4**. This suggests that sulfur would be better than the NH group as a linker. Based on this information, only thioether-type molecules were considered in the designed derivatives. Furthermore, an intermediate-sized polyhedron is located on the benzene ring. Therefore, exploring a pyridine ring as a hydrogen bond acceptor would be reasonable. The last large magenta polyhedron is at the *para* position of the benzene ring in the sulfur-containing derivatives. Thus, exploring either a 4-pyridine ring or inserting hydrogen bond acceptor atoms into the *para* position could be considered. Considering the comprehensive information from both models, everything points to the *para* position being the most favorable in terms of the volume and electronic nature for the exploration of new derivatives.

### 2.4. Design and Synthesis of New Derivatives

Based on the previously described information, we propose a series of new derivatives in which we have prioritized the following structural characteristics: (a) The use of a thioether as a linker between the benzene and quinone; (b) Insertion of bulky but short substituents into the benzene fragment; (c) Avoidance of the use of long chains on the benzene ring; (d) Evaluation of the presence of substituents in all three positions (*ortho*, *meta*, and *para*); (e) The inclusion of a methyl group in the quinone′s pyridine in all derivatives.

The synthesis of the target compounds was carried out as previously described by our group [23,36]. At the first stage, the tricyclic quinone core (QC) was obtained using a “one-pot” reaction, through the oxidation of 1-(2,5-dihydroxyphenyl)-propan-1-one with silver oxide I at room temperature and subsequent [3 + 3] cyclization with the aminouracil ring, yielding the tricyclic hydroquinone intermediate, which rapidly was oxidized aerobically to the QC. At the second stage, the slow and dropwise addition of the thioaryl derivative generated the regioselective addition to the quinone in C-8 [23,25]. With this strategy, we synthesized in two steps a total of 13 new derivatives (compounds **33** to **45**) with good yields. The synthesis route is shown below (Figure 1).

### 2.5. Antibacterial Activity Evaluation

Subsequently, we evaluated in vitro the antibacterial activity of the target compounds against the *Staphylococcus aureus* methicillin-resistant strain (ATCC^®^ 43300) measuring the minimum inhibitory concentration (MIC) using a microbroth dilution technique. The tests were carried out, following the recommendations of the Clinical and Laboratory Standards Institute (CLSI) [37]. Additionally, the compounds were tested in vitro against the *Staphylococcus aureus* methicillin-susceptible strain (ATCC^®^ 29213), *Enterococcus faecalis* (ATCC^®^ 29212), *Escherichia coli* (ATCC^®^ 25922), and *Pseudomonas aeruginosa* (ATCC^®^ 27853).

The results of these new derivates for antimicrobial activity in vitro are reported in Table 4.

The tests evidenced that the quinone derivates show activity against Gram-positive bacteria, within the range of 2 to 32 µg/mL. The most active compounds of the series have a MIC of 2 µg/mL for MRSA (compounds **33**–**35** and **42**) and MSSA (compounds **33** and **34**). Compound **45** has no antibacterial activity. On the other hand, for *E. faecalis*, eight molecules presented activity of 4 µg/mL. Finally, no activity was observed against Gram-negative bacteria.

### 2.6. QSAR Model Challenge

Considering the above results, the challenge of the designed model using the described results was performed. The biological results and the QSAR predictions are presented in Table 5.

As observed in Table 5, all compounds exhibited activity. The most active compounds in the series were molecules **33**–**35** and **42**. The higher activity of these compounds aligns with the information provided by the QSAR models. These molecules feature bulky atoms in the ortho or para positions. Both compounds **33** and **34** have a halogen atom in the para position, which can act as a hydrogen bond acceptor. The lower activity of compound **42**, which has a chlorine atom in the ortho position, may be due to the smaller size of this halogen compared to the bromine derivative, compound **43**. Interestingly, compounds substituted in the meta position were less active (compounds **36**–**40**). This is consistent with the information provided by the CoMSIA model, which discouraged the use of substituents in the meta position of the benzene ring. In the following graph (Figure 5), the distribution of predicted activity values by CoMFA and CoMSIA for each compound is shown. As observed in the graph, all synthesized compounds except for **41** had very good predictions. Compound **41** had a residual value greater than one logarithmic unit in both CoMFA and CoMSIA. Compound **41** features a chlorine atom in the ortho position, while the other derivatives had substituents like bromine, methoxy, and methyl. Therefore, the reason for the observed deviation must go beyond steric reasons. Possibly, the higher electronegativity of the chlorine atom translates into some repulsive interaction with an electron-rich residue in the bacterial target.

## 3. Materials and Methods

### 3.1. QSAR Studies

The CoMFA and CoMSIA studies were executed in the Sybyl X software, version 1.2. The compounds were drawn in ChemDraw and their geometries were relaxed using the MM2 force field. Following this, every compound in mol2 format was minimized using the Powell protocol in Sybyl. The compounds were automatically aligned using the distill rigid alignment. The common quinone nucleus was used as a template for alignment.

The generation of the CoMFA and CoMSIA fields was carried out following the same protocol previously reported by our group [38]. The compounds were manually and randomly divided into training (22 compounds, 70%) and test sets (10 compounds, 30%). The biological activities of each compound were converted into a molar scale prior to the formulation of the models. Each MIC value (mol/L) was converted into pMIC = −logMIC and used as the dependent variable. PLS analysis was used to construct a linear correlation between the CoMFA and CoMSIA descriptors (independent variables) and the activity values (dependent variables). In order to select the final models, the leave-one-out (LOO) method was used to generate the cross-validation coefficient (q^2^) and the optimum number of latent variables (N). The non-cross validation analysis was carried out with a column filter value of 2.0. The QSAR with the highest q^2^ value were selected as the final models: CoMFA-S and CoMSIA-SEA.

### 3.2. Chemistry

The compounds were synthesized using commercial precursors purchased from Sigma-Aldrich^®^ (St. Louis, MO, USA) and benzenethiol from Merck^®^ (Kenilworth, NJ, USA), and were used without purification. All solvents were reagent-grade and readily accessible on the market, and they were utilized without additional purification. TLC aluminum foil 60 F254 (Merck, Darmstadt, Germany) and silica gel (70–230 and 230–400 mesh) were used for the analytical TLC and preparative column chromatography, respectively. ^1^H-NMR spectra (400 MHz) were obtained using AM-400 instruments (Bruker, Billerica, MA, USA) in deuterochloroform (CDCl_3_). The ^13^C-NMR spectra were obtained in CDCl_3_ at 100 MHz. The coupling constants (*J*) are provided in Hertz, and the chemical shift assignments are represented in ppm downfield relative to tetramethylsilane (TMS, δ scale). Using a mass spectrometer equipped with a flight time analyzer (TOF) and a Triwave^®^ system model SYNAPTTM G2, the high-resolution mass spectra (H-RMS) were acquired. (WATERS, Milford, MA, USA). Atmospheric pressure ionization with an electro spray (ESI +/−), a source temperature of 100 °C, a capillarity of 3.0, and a desolvation temperature of 500 °C was used. The melting points (mp) were determined using a Stuart Scientific SMP3 apparatus and were uncorrected.

### 3.3. Chemical Synthesis and Structural Characterization for Compounds

#### 3.3.1. Synthesis of 2,4,6-Trimethylpyrimido[4,5-c]isoquinoline-1,3,7,10(2H,4H)-tetraone (Quinone Core, QC)

A suspension of 1-(2,5-dihydroxyphenyl)-propan-1-one (152.2 mg; 1 mmol), 6-amino-1,3-dimethyl-2,4(1*H*,3*H*)-pyrimidinedione (2) (155.2 mg; 1mmol), Ag_2_O (695.2 mg; 3 mmol), and anhydrous MgSO_4_ (361.1 mg; 3 mmol), was stirred vigorously for three hours at room temperature in dichloromethane (40 mL). The crude was washed and filtered through celite using dichloromethane. The solvent was evaporated under a vacuum, and the crude reaction was purified using 65 g of silica gel (230–400 mesh) column chromatography. A mixture of dichloromethane and ethyl acetate 9:1 was used as eluent. Yellow solid; mp 197.5–198.5 °C (d); ^1^H-NMR (400 MHz; CDCl_3_) δ 7.13 (d, ^3^*J* = 10.5 Hz, 1H, 9-H), 6.83 (d, ^3^*J* = 10.5 Hz, 1H, 8-H), 3.75 (s, 3H, 2-*N*CH_3_), 3.47 (s, 3H, 4-*N*CH_3_), 2.99 (s, 3H, 6-CH_3_). ^13^C-NMR (100 MHz; CDCl_3_) δ 184.2, 183.4, 166.2, 158.3, 152.3, 150.9, 145.8, 138.4, 138.1, 121.1, 105.2, 30.1, 28.9, 26.6; HRMS *m*/*z* 286.0828 (Calculated for C_14_H_12_N_3_O_4_ [M + H]^+^: 286.0832); purified using column chromatography and dichloromethane:ethyl acetate = 9:1; yield: 84% [23,25].

#### 3.3.2. General Procedure (A) for Synthesis of 8-Thioaryl-pyrimidoisoquinolinequinones Derivatives (**33**–**45**)

A solution of benzenethiol derivate (0.5 equiv.) in ethanol: dichloromethane = 1:1 (30 mL) was added dropwise to a solution of QC (150 mg, 0.4909 mmol 1.0 equiv.) and CeCl_3_7H_2_O (5% mmol relative to QC). The reaction mixture was stirred for 16 h at room temperature. Thin-layer chromatography (TLC) was utilized to monitor the reaction progress. The reaction mixture was concentrated under a vacuum, and the crude product was purified using column chromatography (65 g of silica gel 70–230 mesh). The column was eluted with a gradient of petroleum ether, dichloromethane, and ethyl acetate.

##### 8-((4-Chloro-phenyl)thio)-2,4,6-trimethylpyrimido[4,5-c]isoquinoline-1,3,7,10(2H,4H)-tetraone (**33**)

Prepared from QC and 4-chlorothiophenol using general procedure A. Yellow solid; mp 195.3–197.3 °C; ^1^H-NMR (400 MHz; CDCl_3_) δ 7.46–7.51 (m, 4H, 2′-H, 3′-H, 4′-H and 5′-H), 6.19 (s, 1H, 9-H), 3.74 (s, 3H, 2-*N*CH_3_), 3.44 (s, 3H, 4-*N*CH_3_), 3.02 (s, 3H, 6-CH_3_). ^13^C-NMR (100 MHz; CDCl_3_) δ 181.0, 180.7, 166.3, 158.3, 155.9, 152.7, 151.0, 146.8, 137.5, 137.0 (2C), 130.8 (2C), 128.2, 126.7, 125.6, 120.9, 105.7, 30.3, 29.1, 26.9. HRMS *m*/*z* 428.0477 (Calculated for C_20_H_15_ClN_3_O_4_S [M + H]^+^: 428.0472); purified using column chromatography and dichloromethane: ethyl acetate:petroleum ether = 2:2:5; yield: 82%.

##### 8-((4-Bromo-phenyl)thio)-2,4,6-trimethylpyrimido[4,5-c]isoquinoline-1,3,7,10(2H,4H)-tetraone (**34**)

Prepared from QC and 4-bromothiophenol using general procedure A; orange solid; mp 194.4–195.9 °C; ^1^H-NMR (400 MHz; CDCl_3_) δ 7.65 (d, ^3^*J* = 8.4 Hz, 2H, 3′-H and 5′-H), 7.41 (d, ^3^*J* = 8.4 Hz, 2H, 2′-H and 6′-H), 6.19 (s, 1H, 9-H), 3.74 (s, 3H, 2-*N*CH_3_), 3.44 (s, 3H, 4-*N*CH_3_), 3.01 (s, 3H, 6-CH_3_). ^13^C-NMR (100 MHz; CDCl_3_) δ 181.0, 180.7, 166.4, 158.3, 155.8, 152.8, 151.0, 146.8, 137.2 (2C), 133.8 (2C), 128.2, 126.3, 125.7, 120.8, 105.8, 30.3, 29.1, 26.9. HRMS *m*/*z* 471.9964 (Calculated for C_20_H_15_BrN_3_O_4_S [M + H]^+^: 471.9967); purified in column chromatography using dichloromethane: ethyl acetate: petroleum ether = 5:2:2; yield: 80%.

##### 2,4,6-Trimethyl-8-((4-methyl-phenyl)thio)pyrimido[4,5-c]isoquinoline-1,3,7,10(2H,4H)-tetraone (**35**)

Prepared from QC and 4-methylbenzenethiol using general procedure A; orange solid; mp 188.5–191.2 °C; ^1^H-NMR (400 MHz; CDCl_3_) δ 7.40 (d, ^3^*J* = 8.0 Hz, 2H, 2′-H and 6′-H), 7.30 (d, ^3^*J* = 8.0 Hz, 2H, 3′-H and 5′-H), 6.17 (s, 1H, 9-H), 3.73 (s, 3H, 2-*N*CH_3_), 3.43 (s, 3H, 4-*N*CH_3_), 3.00 (s, 3H, 6-CH_3_), 2.42 (s, 3H, 4′-CH_3_). ^13^C-NMR (100 MHz; CDCl_3_) δ 181.2, 181.0, 166.3, 158.4, 156.9, 152.7, 151.0, 147.0, 141.3, 135.5 (2C), 130.7, 131.2 (2C), 128.0, 123.1, 120.9, 105.8, 30.2, 29.1, 26.9, 21.4. HRMS *m*/*z* 408.1018 (Calculated for C_21_H_18_N_3_O_4_S [M + H]^+^: 408.1018); purified using column chromatography and dichloromethane:ethyl acetate:petroleum ether = 9:1:6; yield: 65%.

##### 8-((3-Chloro-phenyl)thio)-2,4,6-trimethylpyrimido[4,5-c]isoquinoline-1,3,7,10(2H,4H)-tetraone (**36**)

Prepared from QC and 3-chlorothiophenol using general procedure A; yellow solid; mp 160.2–162.5 °C; ^1^H-NMR (400 MHz; CDCl_3_) δ 7.49 (m, 4H, 2′-H, 4′-H, 5′-H and 6′-H), 6,20 (s, 1H, 9-H), 3.74 (s, 3H, 2-*N*CH_3_), 3,44 (s, 3H, 4-*N*CH_3_), 3,01 (s, 3H, 6-CH_3_). ^13^C-NMR (100 MHz; CDCl_3_) δ 181.0, 180.6, 166.4, 158.3, 155.6, 152.7, 151.0, 146.8, 136.0, 135.4, 133.8, 131.4, 131.0, 129.0, 128.3, 120.8, 105.7, 30.2, 29.1, 26.9. HRMS *m*/*z* 428.0468 (Calculated for C_20_H_15_ClN_3_O_4_S [M + H]^+^: 428.0472); purified using column chromatography and dichloromethane:ethyl acetate:petroleum ether = 4:1:4; yield: 83%.

##### 8-((3-Bromo-phenyl)thio)-2,4,6-trimethylpyrimido[4,5-c]isoquinoline-1,3,7,10(2H,4H)-tetraone (**37**)

Prepared from QC and 2-bromothiophenol using general procedure A; yellow solid; mp 137.5–139 °C; ^1^H-NMR (400 MHz; CDCl_3_) δ 7.71 (s, 1H, 2′-H), 7.67 (d, ^3^*J* = 7.8 Hz, 1H, 4′-H), 7.49 (d, ^3^*J* = 7.8 Hz, 1H, 6′-H), 7.40 (t, ^3^*J* = 7.9 Hz, 1H, 5′-H), 6.22 (s, 1H, 9-H), 3.75 (s, 3H, 2-*N*CH_3_), 3.45 (s, 3H, 4-*N*CH_3_), 3.02 (s, 3H, 6-CH_3_). ^13^C-NMR (100 MHz; CDCl_3_) δ 181.1, 180.6, 166.4, 158.3, 155.6, 152.8, 151.0, 138.2 (2C), 134.3 (2C), 134.0 131.7, 129.3, 128.3, 123.9, 120.8, 30.3, 29.1, 26.9. HRMS *m*/*z* 471.9956 (Calculated for C_20_H_15_BrN_3_O_4_S [M+H]^+^: 471.9967); purified using column chromatography and dichloromethane:ethyl acetate:petroleum ether = 2:1:6; yield: 64%.

##### 8-((3-Fluoro-phenyl)thio)-2,4,6-trimethylpyrimido[4,5-c]isoquinoline-1,3,7,10(2H,4H)-tetraone (**38**)

Prepared from QC and 3-fluorothiophenol using general procedure A; yellow solid; mp 170.1–172.5 °C; ^1^H-NMR (400 MHz; CDCl_3_) δ 7.47–7.55 (m, 1H, 2′-H), 7.36–7.21 (m, 3H, 4′-H, 5′-H and 6′-H), 6.22 (s, 1H, 9-H), 3.75 (s, 3H, 2-*N*CH_3_), 3.44 (s, 3H, 4-*N*CH_3_), 3.02 (s, 3H, 6-CH_3_). ^13^C-NMR (100 MHz; CDCl_3_) δ 181.1, 180.7, 170.8, 162.9 (d, 1C, ^1^*J* = 251.5 Hz), 158.3, 155.5, 152.8, 151.1, 146.9, 132.0 (d, 1C, ^3^*J* = 8.0 Hz), 131.8 (d, 1C, ^4^*J* = 3.2 Hz), 129.4 (d, 1C, ^3^*J* = 7.6 Hz), 128.1, 122.9 (d, 1C, ^2^*J* = 22.1 Hz), 120.9, 118.3 (d, 1C, ^1^*J* = 20.8 Hz), 105.8, 30.2, 29.1, 26.9. HRMS *m*/*z* 412.0761 (Calculated for C_20_H_15_FN_3_O_4_S [M + H]^+^: 412.0767); purified using column chromatography and dichloromethane:ethyl acetate:petroleum ether = 1:1:6; yield: 88%.

##### 8-(3-Methoxy-phenyl)thio)-2,4,6-trimethylpyrimido[4,5-c]isoquinoline-1,3,7,10(2H,4H)-tetraone (**39**)

Prepared from QC and 2-methoxythiophenol using general procedure A; yellow solid; mp 177.1–178.6 °C (d); ^1^H-NMR (400 MHz; CDCl_3_) δ 7.41 (t, ^3^*J* = 8.0 Hz, 1H, 5′-H), 7.11 (d, ^3^*J* = 7.7 Hz, 1H, 6′-H), 7.06 (s, 1H, 2′-H), 7.05 (d, ^3^*J* = 8.0 Hz, 1H, 4′-H), 6.23 (s, 1H, 9-H), 3.84 (s, 3H, 3′-CH_3_), 3.74 (s, 3H, 2-*N*CH_3_), 3.43 (s, 3H, 4-*N*CH_3_), 3.01 (s, 3H, 6-CH_3_). ^13^C-NMR (100 MHz; CDCl_3_) δ 181.2, 180.9, 166.3, 160.8, 158.3, 156.4, 152.7, 151.0, 146.9, 131.3, 128.1, 128.0, 127.7, 120.9, 120.7, 116.6, 105.7, 55.5, 30.2, 29.1, 26.9. HRMS *m*/*z* 424.0976 (Calculated for C_21_H_18_N_3_O_5_S [M + H]^+^: 424.0967); purified using column chromatography and dichloromethane:ethyl acetate:petroleum ether = 9:1:6; yield: 80%.

##### 2,4,6-Trimethyl-8-((3-methyl-phenyl)thio)pyrimido[4,5-c]isoquinoline-1,3,7,10(2H,4H)-tetraone (**40**)

Prepared from QC and 3-methylbenzenethiol using general procedure A; orange solid; mp 158.3–159.9 °C; ^1^H-NMR (400 MHz; CDCl_3_) δ 7.49 (d, ^3^*J* = 7.5 Hz, 1H, 6′-H), 7.39–7.45 (m, 2H, 2′-H and 5′-H), 7.31 (t, ^3^*J* = 7.5 Hz, 1H, 4′-H), 6.01 (s, 1H, 9-H), 3.74 (s, 3H, 2-*N*CH_3_), 3.43 (s, 3H, 4-*N*CH_3_), 3.02 (s, 3H, 6-CH_3_), 2.42 (s, 3H, 3′-CH_3_). ^13^C-NMR (100 MHz; CDCl_3_) δ 181.1 (2C), 166.3, 158.4, 155.3, 152.7, 151.0, 147.0, 143.1, 136.7, 133.7, 133.3, 127.9, 127.6, 126.2, 120.3, 106.7, 30.2, 29.1, 27.0, 20.5. HRMS *m*/*z* 408.1012 (Calculated for C_21_H_18_N_3_O_4_S [M + H]^+^: 408.1018); purified using column chromatography and dichloromethane:ethyl acetate:petroleum ether = 2:1:6; yield: 65%.

##### 8-((2-Chloro-phenyl)thio)-2,4,6-trimethylpyrimido[4,5-c]isoquinoline-1,3,7,10(2H,4H)-tetraone (**41**)

Prepared from QC and 2-chlorothiophenol using general procedure A; yellow solid; mp 208.5 °C (d); ^1^H-NMR (400 MHz; CDCl_3_) δ 7.64 (m, 2H, 3′-H and 6′-H), 7.40 (m, 2H, 4′-H and 5′-H), 5.87 (s, 1H, 9-H), 3.74 (s, 3H, 2-*N*CH_3_), 3.48 (s, 3H, 4-*N*CH_3_), 2.95 (s, 3H, 6-CH_3_). ^13^C-NMR (100 MHz; CDCl_3_) δ 181.9, 180.6, 166.2, 158.3, 153.6, 152.0, 151.0, 145.8, 139.8, 137.8, 132.4, 131.2, 128.4, 127.9, 126.0, 121.3, 105.5, 30.2, 29.0, 26.7. HRMS *m*/*z* 428.0468 (Calculated for C_20_H_15_ClN_3_O_4_S [M + H]^+^: 428.0472); purified using column chromatography and dichloromethane:ethyl acetate:petroleum ether = 3:1:4; yield: 88%.

##### 8-((2-Bromo-phenyl)thio)-2,4,6-trimethylpyrimido[4,5-c]isoquinoline-1,3,7,10(2H,4H)-tetraone (**42**)

Prepared from QC and 2-bromothiophenol using general procedure A; yellow solid; mp 210.7 °C (d); ^1^H-NMR (400 MHz; CDCl_3_) δ 7.81 (d, ^3^*J* = 7.5 Hz, 1H, 3′-H), 7.66 (d, ^3^*J* = 7.7 Hz, 1H, 6′-H), 7.42 (m, 2H, 4′-H and 5′-H), 6.06 (s, 1H, 9-H), 3.74 (s, 3H, 2-*N*CH_3_), 3.44 (s, 3H, 4-*N*CH_3_), 3.03 (s, 3H, 6-CH_3_). ^13^C-NMR (100 MHz; CDCl_3_) δ 180.9, 180.7, 166.3, 158.3, 153.8, 152.8, 151.0, 146.8, 137.9, 134.6, 132.4, 130.7, 129.2, 128.7, 128.1, 120.9, 105.8, 30.2, 29.0, 26.9. HRMS *m*/*z* 471.9968 (Calculated for C_20_H_15_BrN_3_O_4_S [M + H]^+^: 471.9967); purified using column chromatography and dichloromethane:ethyl acetate:petroleum ether = 15:3:5; yield: 97%.

##### 8-(2-Methoxy-phenyl)thio)-2,4,6-trimethylpyrimido[4,5-c]isoquinoline-1,3,7,10(2H,4H)-tetraone (**43**)

Prepared from QC and 2-methoxythiophenol using general procedure A; orange solid; mp 170.2 °C (d); ^1^H-NMR (400 MHz; CDCl_3_) δ 7.52 (t, ^3^*J* = 7.8 Hz, 2H, 4′-H and 6′-H), 7.06 (t, ^3^*J* = 7.5 Hz, 1H, 3′-H and 5′-H), 6.11 (s, 1H, 9-H), 3.86 (s, 3H, 2-OCH_3_), 3.74 (s, 3H, 2-*N*CH_3_), 3.43 (s, 3H, 4-*N*CH_3_), 3.01 (s, 3H, 6-CH_3_). ^13^C-NMR (100 MHz; CDCl_3_) δ 181.2, 181.1, 166.2, 160.0, 158.4, 154.5, 152.6, 151.0, 147.0, 137.5, 133.0, 127.7, 122.0, 121.2, 114.4, 112.0, 105.7, 56.1, 30.2, 29.0, 26.9. HRMS *m*/*z* 424.0958 (Calculated for C_21_H_18_N_3_O_5_S [M + H]^+^: 424.0967); purified using column chromatography and dichloromethane:ethyl acetate: petroleum ether = 9:1:6; yield: 93%.

##### 2,4,6-Trimethyl-8-((3-methyl-phenyl)thio)pyrimido[4,5-c]isoquinoline-1,3,7,10(2H,4H)-tetraone (**44**)

Prepared from QC and 2-methylbenzenethiol using general procedure A; yellow solid; mp 204.5.0–205.8 °C; ^1^H-NMR (400 MHz; CDCl_3_) δ 7.38 (d, ^3^*J* = 7.2 Hz, 1H, 6′-H), 7.34–7.30 (m, 3H, 3′, 4′ and 5′-H), 6.19 (s, 1H, 9-H), 3.74 (s, 3H, 2-*N*CH_3_), 3.43 (s, 3H, 4-*N*CH_3_), 3.03 (s, 3H, 6-CH_3_), 2.41 (s, 3H, 2′-CH_3_). ^13^C-NMR (100 MHz; CDCl_3_) δ 181.4, 181.0, 166.4, 158.5, 156.8, 152.8, 151.1, 147.1, 140.7, 136.3, 132.7, 131.8, 130.4, 128.2, 126.8, 121.1, 105.8, 30.4, 29.1, 27.1, 21.4. HRMS *m*/*z* 408.1011 (Calculated for C_21_H_18_N_3_O_4_S [M + H]^+^: 408.1018); purified using column chromatography and dichloromethane:ethyl acetate:petroleum ether = 2:1:6; yield: 68%.

##### 8-((2-Fluoro-phenyl)thio)-2,4,6-trimethylpyrimido[4,5-c]isoquinoline-1,3,7,10(2H,4H)-tetraone (**45**)

Prepared from QC and 2-fluorothiophenol using general procedure A; yellow solid; mp 216.2 °C (d); ^1^H-NMR (400 MHz; CDCl_3_) δ 7.50–7.56 (m, 2H, 4′-H and 6′-H), 7.22–7.28 (m, 2H, 3′-H and 5′-H), 6.14 (s, 1H, 9-H), 3.71 (s, 3H, 2-*N*CH_3_), 3.40 (s, 3H, 4-*N*CH_3_), 2.98 (s, 3H, 6-CH_3_). ^13^C-NMR (100 MHz; CDCl_3_) δ 181.1, 180.8, 166.3, 162.7 (d, 1C, ^4^*J* = 251.4 Hz), 158.3, 153.8, 152.7, 151.0, 146.8, 137.5, 133.8 (d, 1C, ^1^*J* = 8.0 Hz), 128.2, 125.9 (d, 1C, ^2^*J* = 3.9 Hz), 120.9, 117.2 (d, 1C, ^2^*J* = 22.2 Hz), 114.4 (d, 1C, ^4^*J* = 18.8 Hz), 105.7, 30.3, 29.1, 27.0. HRMS *m*/*z* 412.0768 (Calculated for C_20_H_15_FN_3_O_4_S [M + H]^+^: 412.0767); purified using column chromatography and dichloromethane:ethyl acetate:petroleum ether = 2:2:5; yield: 68%.

### 3.4. Evaluation of Antibacterial Activity

The determination of the antibacterial activity was performed using the microdilution method in culture broth according the procedures of The Clinical and Laboratory Standards Institute (CLSI) [37]. The minimum inhibitory concentration (MIC) for each bacterial type was determined. The following bacteria were used in the evaluations: *Staphylococcus aureus* methicillin-susceptible strain (ATCC^®^ 43300), *Staphylococcus aureus* methicillin-susceptible strain (ATCC^®^ 29213), *Enterococcus faecalis* (ATCC^®^ 29212), *Escherichia coli* (ATCC^®^ 25922), and *Pseudomonas aeruginosa* (ATCC^®^ 25923). Dimethyl sulfoxide (DMSO, maximum of 1% per well) was used to dissolve each drug tested. As a quality control measure, the data were compared with the MIC ranges given by the CLSI using vancomycin and gentamicin as references against the strains [37]. In addition, bacterial growth controls and broth sterility controls were used as quality controls for the assay. The maximum concentration for the compounds and standard drugs was 32 μg/mL. The inoculum was prepared to a final concentration of 5 × 10^5^ CFU/mL in the test tray. The plates were incubated at 35 °C for 18–20 h. All experiments were performed in triplicate.

## 4. Conclusions

The formulated models demonstrated that steric, electronic, and hydrogen-bond acceptor properties contribute to the biological activity of the studied compounds. Both models exhibited good statistical values of q^2^ (0.660 and 0.596) and r^2^ (0.938 and 0.895). Additionally, the Y-randomization test showed that the results are not the result of random correlation. The main structure–activity relationships found were that short, bulky, electron-rich groups with a hydrogen bond acceptor capability on the benzene ring are favorable for antibacterial activity. Based on this information, a series of 13 new compounds were synthesized via two synthesis stages with good yields. There were 12 molecules that presented antibacterial activity against Gram-positive bacteria in a range of 2 to 32 µg/mL. Finally, of the total number of synthesized compounds, only one did not align well with the predictions.

## 5. Patents

PatentWO2017113031A1, USAUS11390622B2, PCT/CL2015003780A1, EPO EP3404026A4; China CN109121411B. MX/a/2018/008192A titled: “Pyrimidine-Isoquinoline-Quinone Derived Compounds, their Salts, Isomers, Pharmaceutically Acceptable Tautomers; Pharmaceutical Composition; Preparation Procedure; and their Use in the Treatment of Bacterial and Multi-Resistant Bacterial Diseases”.

## Data Availability

Information on the chemical structures used in the formulation of the models can be found in the supplementary material.

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
