# Peer review of "QSAR Studies, Synthesis, and Biological Evaluation of New Pyrimido-Isoquinolin-Quinone Derivatives against Methicillin-Resistant Staphylococcus aureus"

_pharmaceuticals, 2023, doi:10.3390/ph16111621_

Round 1

Reviewer 1 Report

Comments and Suggestions for Authors

The manuscript describes the synthesis of 13 new pyrimidoisoquinolinquinones based compounds designed by computational approach. The antibacterial activity of the synthesized compounds were evaluated and out of the 13 synthesized compounds, 12 were found to possess promising activity profile. The manuscript is interesting and will be useful for the scientific community, especially for researchers working in the area of drug discovery. The research design is appropriate and the the conclusions presented are supported by the results. The references are also found to be appropriate. 

The manuscript can be accepted for publication after major revision. The comments are detailed below:

1) Is there any supporting information uploaded along with this manuscript? I could not find any supporting information (with spectra of new compounds) in the system.

2) Figure 5 should be actually Scheme 1. Also the time taken for the reaction needs to be included for the first step.

3) Table 4, the number of replicates should be included in footnote.

4) Line 357-362, present the synthetic procedure in a readable form and also in past tense. Also mention the compounds as "yellow solid", orange solid etc.

5) Line 379, what is light petroleum?

6) The manuscript needs to be rephrased at many instances to make it interesting to the readers. This manuscript should be thoroughly proofread before publication. 

Comments on the Quality of English Language

Moderate corrections are required.

Reviewer 2 Report

Comments and Suggestions for Authors

This is an excellent contribution that describes the use of QSAR studies to design, synthesize and evaluate a series of heterocyclic compounds against an important pathogen.  The manuscript is generally  well written, although improvements are needed for paragraph structure.  The figures and tables are excellent.  The main issue to be addressed is the QSAR models are built based on w/v activities and not molar values.  As a results, the QSAR equations predict activity weighted by molecular weight.  This is not uncommon, but the issues and limitations of model interpretation should be discussed.  The conclusion are supported by the results, and the materials and methods sections are written in detail such that the experiments can  be reproduced.   The characterization of the compounds is excellent.  Overall, this is an excellent contribution that has a few minor issues that should be addressed within the manuscript prior to publication.

Specific Comments

 The paragraph structure of the submission can be improved.  The first paragraph of the introduction is 1 sentence.  Paragraphs should be at least 3+ paragraphs.  The authors have an opportunity to expand their opening point.

1.            Most of the paragraphs in this manuscript suffer from the same issue.  These paragraphs should be combined or the points expanded upon. 

2.            The introduction can benefit from a review of the merits and limitations of CoMFA and ComSIA.

3.            The q2= cross-validation coefficient is commonly used to evaluate QSAR model predictability, and the scores are at the borderline of acceptability.  The authors may wish to discuss the merits and limitations of the use of models with borderline acceptable

4.            The Mean IC50 is given in ug/ml.  The compounds have different molecular weights. Typically, biological activities for QSAR studies are given in mol/L (micromole/L) so that the activity of each compound is equally weighted based on molecular weight. A comment on the importance of molar values vs w/volume values can benefit readers.

5.            The authors my wish to rerun the QSAR models after converting the biologically activities to mmol/L or an equivalent molar value.

6.            The  merits and limitations of building QSAR models using ug/mL values could be discussed.  For example – the models presented predict a composite of molecular weight and biological activity.  QSAR models based on molar values predict biological activity.

7.            Figure 6 is excellent.

8.            The results/discussion, experiment, and conclusion sections are excellent.
